# STKAN: KOLMOGOROV-ARNOLD NETWORKS FOR SPATIO-TEMPORAL FORECASTING

## ABSTRACT

Real-world traffic data exhibit intricate, intertwined spatial and temporal dynamics, significantly complicating accurate forecasting. Recent decomposition-based approaches aim to disentangle these complex dynamics into separate spatial and temporal components, facilitating clearer and more effective modeling. However, varying information densities between spatial structures and temporal patterns remain a substantial challenge, potentially leading to inaccurate feature interactions and subsequently degraded forecast performance. To address these challenges, inspired by Kolmogorov–Arnold Networks (KANs), we propose a novel Spatio-Temporal Decomposition Learning architecture (STKAN). The STKAN framework explicitly separates and individually models spatial and temporal dependencies using specialized multi-order KAN modules. It encodes complex input series into spatio-temporal embeddings through an adaptive node–group assignment mechanism. Dedicated spatial and temporal KAN modules independently and robustly capture inter-node relationships and temporal dynamics at multiple orders, thereby modeling distinct underlying patterns more effectively. Extensive experimental evaluations on widely recognized benchmark datasets convincingly demonstrate that STKAN achieves state-of-the-art forecasting accuracy, while maintaining scalability and robustness across diverse traffic scenarios. In particular, STKAN consistently adapts to networks of varying sizes and traffic regimes without the need for architecture-specific tuning. Moreover, its decomposition design provides a principled way to balance model complexity with learning stability, making it well-suited for real-world deployment. Code will be released upon notification.

## 1 INTRODUCTION

In recent years, traffic time series data collected from road sensors have emerged as a crucial research focus in the field of intelligent transportation Chen et al. (2018); Wang et al. (2022); Lin et al. (2022a). Predicting future road traffic conditions based on historical data plays a vital role in many real-world intelligent transportation applications Wang et al. (2020); Deng et al. (2024). Accurate traffic forecasting is particularly valuable for alleviating urban congestion, reducing carbon emissions, and improving road safety Han et al. (2022); Lin et al. (2022b). However, traffic data inherently exhibit strong temporal autocorrelations and spatial interdependencies, which makes it highly challenging to effectively capture such complex spatio-temporal relationships.

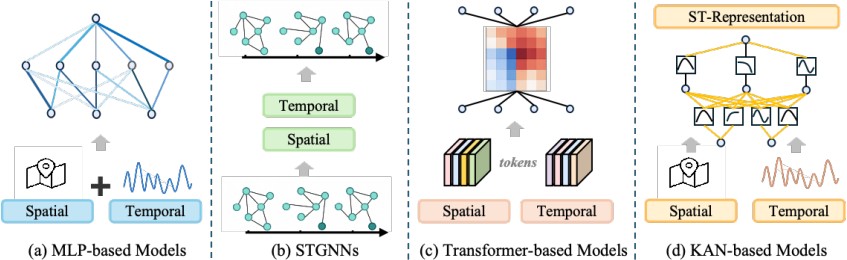

**Figure 1:** Illustration of interpretability mechanisms in different spatio-temporal forecasting models.

To disentangle the intertwined spatial and temporal dependencies in traffic data, recent approaches often adopt decomposition strategies that convert raw signals into spatial and temporal components for clearer modeling. Hierarchical methods such as STGCN Han et al. (2020) alternate temporal convolutions with graph convolutions, progressively isolating temporal dynamics from spatial interactions at each layer. Dynamic topology learning models, including Graph WaveNet Wu et al. (2019), DGCRN Li et al. (2023), and MegaCRN Jiang et al. (2023), leverage hypernetworks to adaptively generate adjacency matrices, dynamically conditioning spatial structures within temporal contexts. Multiscale spectral decomposition techniques, such as StemGNN Cao et al. (2020) and STWave Fang et al. (2023), apply Fourier or wavelet transforms to split traffic signals into frequency-based subseries, enabling independent modeling of short-term fluctuations and long-term trends. Representation-level frameworks, such as STID Shao et al. (2022), introduce identity-based embeddings that explicitly factorize node-specific spatial characteristics from shared temporal patterns. Collectively, these decomposition-based methods aim to obtain latent spatio-temporal representations that support more accurate predictions.

Nevertheless, simply concatenating spatial and temporal features, or fusing them with fixed nonlinearities, often fails to reconcile the different information densities and statistical properties of the two dimensions. This limitation reduces representational capacity and leads to suboptimal optimization. Existing MLP-based models are particularly constrained, as they rely on fixed activation functions whose forms remain unchanged during training. Such rigidity prevents them from fully capturing the diverse and complex nonlinear spatio-temporal dependencies present in real traffic data. By contrast, GNN-based methods perform spatio-temporal forecasting by assuming a predefined or dynamically learned graph structure to capture spatial correlations among nodes. Transformer-based methods, on the other hand, split fused spatio-temporal sequences into tokens and apply attention to model dependencies across them, as illustrated in Figure 1.

However, the complex structures of existing spatio-temporal networks still make it difficult to explicitly disentangle spatial and temporal dependencies. Most models process spatio-temporal signals in a unified manner, without a clear mechanism to extract and separate spatial and temporal features. This often leads to mixed representations where spatial interactions and temporal dynamics are not explicitly distinguished. Therefore, a critical question arises: can we design a framework that decomposes and models spatial and temporal patterns separately, thereby improving representation quality and predictive performance? Recently proposed Kolmogorov–Arnold Networks (KANs) provide a promising direction, as they support flexible kernel choices and adjustable orders, allowing adaptive representation of spatial and temporal dependencies at different levels.

Motivated by these limitations, we propose STKAN, a Kolmogorov–Arnold Network (KAN)-based architecture that explicitly decomposes and models spatial and temporal dependencies. STKAN introduces adaptive node–group assignments to form spatio-temporal embeddings, followed by specialized spatial and temporal multi-order KAN modules that independently capture inter-node correlations and temporal dynamics. The resulting representations are then fused for accurate forecasting. This explicit decomposition enables the model to avoid the entanglement of heterogeneous patterns that often occurs in existing architectures. In addition, the flexibility of KAN allows STKAN to adaptively balance modeling capacity and optimization stability, making it effective for spatio-temporal forecasting.

Our contributions are summarized as follows:

- We revisit spatio-temporal forecasting by introducing **STKAN**, a decomposition-based framework that separates and then fuses spatial and temporal components, enabling clearer representations and more accurate modeling.

- We design dedicated **spatial and temporal KAN blocks**, where node–group assignments highlight spatial interactions and multi-order expansions capture temporal dynamics across scales, improving adaptability and stability.

- We validate STKAN on benchmark traffic datasets, showing consistent gains in prediction accuracy, scalability, and robustness over state-of-the-art methods, without requiring task-specific architectural customization.

## 2 RELATED WORK

### 2.1 SPATIO-TEMPORAL FORECASTING

Spatial-temporal forecasting extends traditional time-series forecasting by incorporating both temporal dynamics and spatial dependencies, such as in traffic management, where multiple traffic sensors' data is used to predict future conditions. Early deep learning approaches combined Convolutional Neural Networks (CNNs) and Recurrent Neural Networks (RNNs) to capture spatial and temporal dependencies Shi et al. (2015); Yao et al. (2018); Lai et al. (2018). However, grid-based CNNs may not effectively handle non-Euclidean spatial relationships, leading to the development of Graph Convolutional Networks (GCNs) Defferrard et al. (2016); Kipf & Welling (2016) and Spatial-Temporal Graph Neural Networks (STGNNs) Li et al. (2017); Yu et al. (2017). These models, such as DCRNN Li et al. (2017), ST-MetaNet Pan et al. (2019), and DGCRN Li et al. (2023), integrate GCNs with RNNs Cho et al. (2014), while others like Graph WaveNet Wu et al. (2019) and STGCN Yu et al. (2017) combine GCNs with gated Temporal Convolutional Networks (TCNs). Attention mechanisms have also been widely adopted in STGNNs Zheng et al. (2020). Despite their success, some studies criticize the reliance on pre-defined graphs, suggesting alternatives like AGCRN Bai et al. (2020) and MTGNN Wu et al. (2020), which learn latent graph structures. However, both prior and latent graph-based STGNNs often involve high computational complexity. Recent research has proposed more efficient non-GCN solutions, such as STNorm Deng et al. (2021) and STID Shao et al. (2022), which achieve similar performance with greater efficiency, highlighting the need for a better understanding of spatial dependencies in forecasting tasks.

### 2.2 KOLMOGOROV-ARNOLD NETWORK

KANs leverage the Kolmogorov–Arnold theorem Liu et al. (2024), decomposing complex multivariate functions into combinations of simpler univariate functions, enhancing nonlinear modeling capabilities. Recent advancements include Multi-layer Mixture-of-Experts KAN Han et al. (2024), which adaptively selects optimal expert functions, such as wavelet-based WavKAN Bozorgasl & Chen, Taylor polynomial-based TaylorKAN Yu et al. (2025), and Jacobi polynomial-based JacobiKAN Aghaei (2025), significantly improving performance and interpretability for multivariate time series prediction. FastKAN further boosts computational efficiency through Gaussian radial basis functions Li (2024). KANs have seen growing use in time series prediction, with methods like T-KAN and MT-KAN using symbolic regression for interpreting nonlinear temporal patterns Xu et al. (2024). iTFKAN employs collaborative time-frequency learning for robust decision-making Liang et al. (2025), while TimeKAN incorporates cascaded frequency decomposition and higher-order KAN representations to capture complex frequency dynamics effectively Huang et al. (2025). However, the application of KANs to spatio-temporal prediction remains unexplored. This study introduces STKAN, a framework leveraging refined spatio-temporal decoupling to efficiently and interpretably forecast complex spatio-temporal systems using KAN's nonlinear modeling strengths.

## 3 PRELIMINARY

Spatio-temporal forecasting is a specialized multivariate time–series forecasting problem. Given the multivariate time series $X_{t-(T-1):t}$ from the past $T$ time steps, our goal is to predict the next $T$ time steps:

$$\begin{bmatrix} X_{t-(T+1)}, \, \ldots, \, X_t \end{bmatrix} \; \longrightarrow \; \begin{bmatrix} \widehat{Y}_{t+1}, \, \ldots, \, \widehat{Y}_{t+T} \end{bmatrix}$$

where $X_i \in \mathbb{R}^{N \times C}$ denotes the observation at the $i$-th time step, $N$ is the number of spatial nodes, and $C$ is the number of information channels

## 4 METHODOLOGY

In this paper, we propose **STKAN** to effectively capture both symbolic spatial dependencies among nodes and temporal dependencies across time steps. The overall architecture of STKAN is shown in Figure 2.

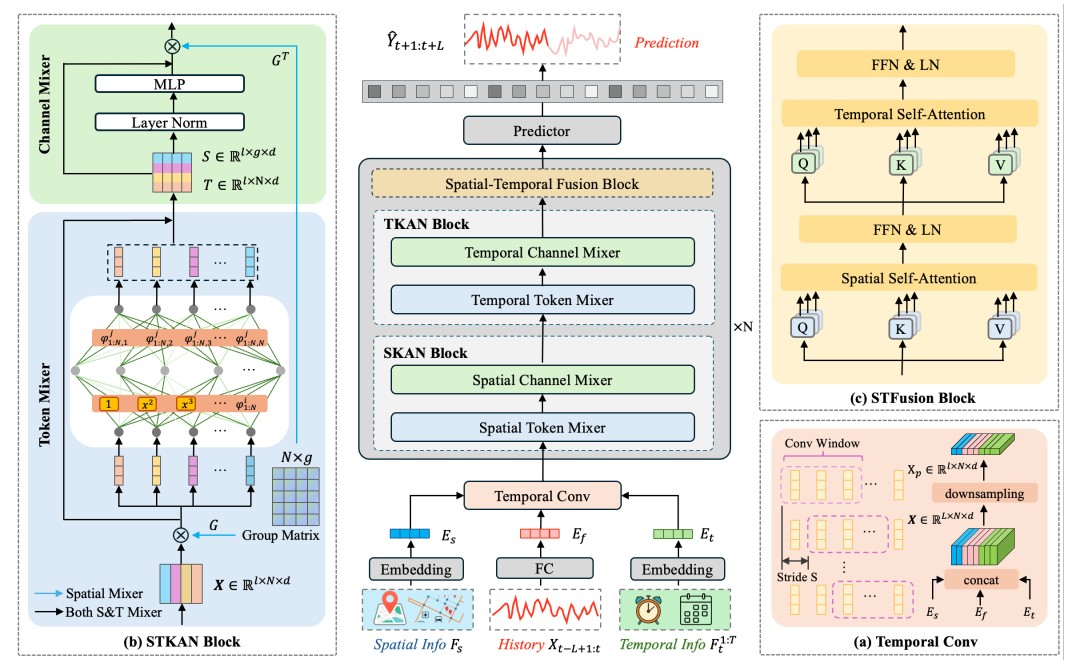

**Figure 2:** Overview of our proposed STKAN framework.

## 4.1 EMBEDDING LAYERS

To capture the complex spatio-temporal dependencies in traffic sequences, we utilize a fully connected layer to embed the raw data into a high-dimensional feature representation. Given a historical input sequence $X_{t-L+1:t} \in \mathbb{R}^{L \times N \times C}$, where $L$ is the sequence length, $N$ is the number of nodes, and $C$ is the number of features, we generate the feature embedding $E_f \in \mathbb{R}^{L \times N \times d_f}$ through $E_f = FC(X_{t-T+1:t})$, with $d_f$ representing the embedding dimension and $FC(\cdot)$ denoting the fully connected layer. We denote a learnable spatial embedding tensor $E_s \in \mathbb{R}^{L \times N \times d_s}$ to capture static spatial characteristics among nodes, shared across all timesteps. Furthermore, to incorporate temporal periodicity, we design two embedding dictionaries: one for time-of-day with embeddings $E_d \in \mathbb{R}^{T_d \times d_d}$ and another for day-of-week with embeddings $E_w \in \mathbb{R}^{T_w \times d_w}$, where $T_d = 7$ and $T_w = 288$ are the number of intervals per day and week, respectively. We obtain the temporal embedding $E_t \in \mathbb{R}^{L \times N \times d_t}$, where $d_t = d_d + d_w$.

## 4.2 TEMPORAL CONVOLUTION BLOCKS

All embeddings are concatenated along the feature dimension to form the final spatio-temporal representation $X \in \mathbb{R}^{L \times N \times d_h}$, defined as

$$X = E_f \parallel E_s \parallel E_t, \tag{1}$$

where the hidden dimension satisfies $d_h = d_f + d_s + d_t$.

To capture local temporal context while shortening the sequence, we define a patch-extraction operator $\mathrm{PatchConv}(\cdot; w, S)$ that applies a $1 \times w$ convolution along the time axis with stride $S$:

$$X_p = Conv\left(X; w, S\right), \tag{2}$$

where $X_p \in \mathbb{R}^{l \times N \times d_h}$, this produces a patch-level representation and the resulting length is $l = \lfloor (L - w)/S \rfloor + 1$.

Thus, $X_p$ serves as a down-sampled spatio-temporal feature map that preserves salient temporal patterns while reducing computational cost in subsequent mixer or attention blocks.

## 4.3 SPATIO-TEMPORAL KAN BLOCKS

To effectively extract information from the hidden spatio-temporal representations generated by the embedding module, we propose the STKAN blocks, which systematically integrates spatial and temporal information to capture the intricate dynamics of spatio-temporal data.

Specifically, STKAN first employs a learnable group matrix to aggregate raw spatial nodes into higher-level spatial representations, enabling the model to capture more coarse-grained spatial patterns. The aggregated representations are then fed into the Spatial KAN (SKAN) blocks, which is designed to model the complex interactions among spatial groups. After spatial refinement, a residual fusion mechanism integrates the spatial outputs with the original input, preserving both global and local information. Finally, the fused representations are passed to the Temporal KAN (TKAN) blocks, which focuses on learning dynamic temporal patterns and dependencies across time steps, enabling a comprehensive understanding of the evolving spatio-temporal processes.

**Learnable Spatial Grouping.** To transform the original spatial nodes into macro-level spatial tokens and to expose how each node contributes to every group for later interpretability, we introduce a learnable spatial adaptive group matrix. Let the node-level feature tensor be $X \in \mathbb{R}^{l \times N \times d}$. A matrix of learnable parameters is normalized in rows by the softmax function to produce a group assignment matrix $G \in \mathbb{R}^{N \times g}$. The resulting group-level representation $\widetilde{X} \in \mathbb{R}^{l \times g \times d}$ is obtained

$$\widetilde{X}_{l,g,d} = \sum_{n=1}^{N} G_{N,g} X_{l,N,d}, \tag{3}$$

where $g$ denotes the number of spatial groups.

**Spatial KAN Block.** Comprehensively modeling spatial dependencies across nodes is inherently challenging due to their complex interrelationships. To address this difficulty, we adopt a mixer-based architecture that enables flexible interactions among spatial tokens. Conventional mixer architectures typically rely on standard MLPs to facilitate token mixing. Such MLPs apply fixed nonlinear activation functions at each node, and once determined, these functions remain unchanged throughout the network. Although this approach has proven effective for numerous tasks, employing fixed nonlinearities may constrain the model's ability to capture highly intricate, nonstandard, or task-specific patterns in the data.

In contrast, the recently proposed Kolmogorov–Arnold Networks (KANs) offer a more flexible alternative. The core idea of KAN is to place learnable activation functions on the edges of the network, rather than using fixed activations at the nodes as in MLPs. This enables each neuron to connect to neurons in the previous layer via a learnable univariate function $\phi$, allowing the network to adapt its nonlinear transformations based on data. The transmission from the $j$-th neuron in layer $l + 1$ to all neurons in layer $l$ is formulated as:

$$z_{l+1,j} = \sum_{i=1}^{n_l} \phi_{l,j,i}(z_{l,i}), \tag{4}$$

where $z_{l,i}$ is the $i$-th neuron in layer $l$, and $n_l$ is the number of neurons in that layer. Here, $\phi_{l,j,i}(\cdot)$ is a learnable univariate mapping, enabling expressive nonlinear modeling with fewer parameters and improved interpretability.

In our model, we instantiate $\phi$ using the **TaylorKANLayer**, which leverages a Taylor series expansion to parameterize the learnable edge activations. Benefiting from its intrinsic sensitivity to short-term changes, this layer naturally captures local variations, making it particularly suitable for modeling fine-grained spatial dependencies among nodes.

Specifically, $\phi(x)$ is approximated by a $k$-th order Taylor series:

$$T_k(x) = \sum_{i=0}^{k} \frac{f^{(i)}(0)}{i!} x^i, \tag{5}$$

where $f^{(i)}(0)$ denotes the $i$-th derivative evaluated at zero. In TaylorKANLayer, instead of fixed derivatives, the coefficients are learned from data, allowing flexible adaptation. A 1-layer Tay-

lorKAN applied to a multi-dimensional input is expressed as:

$$\phi_q(\boldsymbol{x}) = \sum_{p=1}^{C} \sum_{i=0}^{k} \Theta_{q,p,i}(x_p)^i + b_q, \tag{6}$$

$$\text{KAN}(\boldsymbol{x}) = \left\{ \begin{array}{c} \phi_1(\boldsymbol{x}) \\ \vdots \\ \phi_C(\boldsymbol{x}) \end{array} \right\}, \tag{7}$$

where $\boldsymbol{x} \in \mathbb{R}^C$, $\Theta \in \mathbb{R}^{C \times C \times (k+1)}$ are learnable coefficients, and $b_q$ is a bias term. This formulation allows the TaylorKANLayer in the SKAN to capture adaptive and nonlinear interactions across the spatial dimension. Specifically, it approximates the relationships between nodes and their corresponding groups by adaptively combining multiple polynomial terms derived from the Taylor series expansion, enabling precise modeling of complex spatial dependencies. Therefore, the *token-mixing* stage is formulated as:

$$S_t = \widetilde{X}^\top + \text{TaylorKAN}_k(\widetilde{X}^\top; \Theta), \tag{8}$$

where, $S_t \in \mathbb{R}^{l \times d \times g}$

Following the token mixing, the module proceeds to the *channel-mixing* stage, where a standard MLP is applied independently at each spatial group to capture intra-feature interactions. The channel mixer is formulated as:

$$S_c = S_t^\top + W_2\, \sigma\left(W_1\, \text{LayerNorm}(S_t^\top) + b_1\right) + b_2, \tag{9}$$

where $\sigma$ denotes the GELU activation, and $W_1, W_2, b_1, b_2$ are learnable parameters. The final output of SKAN is the sum of the outputs from the token-mixer and channel-mixer.

$$\mathcal{S} = (S_t + S_c)\, G^\top, \tag{10}$$

**Temporal KAN Blocks.** Similar to the SKAN blocks, the core operations of the TKAN focus on the temporal dimension, aiming to capture the dynamic evolution patterns and long-range dependencies inherent in time series data. Unlike spatial mixing, temporal mixing directly operates on the individual time steps of a time series, where each time step itself serves as a temporal token.

$$T_t = S^\top + \text{TaylorKAN}_k(S^\top; \Theta), \tag{11}$$

$$T_c = T_t^\top + W_4\, \sigma\left(W_3\, \text{LayerNorm}(T_t^\top) + b_3\right) + b_4, \tag{12}$$

where $W_3, W_4, b_3, b_4$ are learnable parameters. The final output of TKAN is the sum of the outputs from the token-mixer and channel-mixer.

$$\mathcal{T} = T_t + T_c, \tag{13}$$

### 4.4 Spatio-temporal Fusion Blocks

To complement the local sensitivity of TaylorKAN, we introduce Transformer layers along both spatial and temporal axes. Given a hidden representation $Z \in \mathbb{R}^{L \times N \times d_h}$ (with $L$ time steps and $N$ spatial nodes), we project it into query, key, and value matrices:

$$Q = ZW_Q, \quad K = ZW_K, \quad V = ZW_V, \tag{14}$$

where $W_Q, W_K, W_V \in \mathbb{R}^{d_h \times d_h}$ are learnable parameters. The standard scaled dot-product attention is then applied as

$$\text{Attn}^{(a)} = \text{Softmax}\left(\frac{Q^{(a)} K^{(a)\top}}{\sqrt{d_h}}\right), \qquad Z^{(a)} = \text{Attn}^{(a)} V^{(a)}, \tag{15}$$

with $a \in \{\text{s}, \text{t}\}$ denoting the *spatial* or *temporal* mode.

Specifically, spatial attention is computed *independently at each time step* across nodes, yielding $\text{Attn}^{(s)} \in \mathbb{R}^{T \times N \times N}$. Temporal attention is computed *independently at each node* across time, yielding $\text{Attn}^{(t)} \in \mathbb{R}^{N \times L \times L}$. This design enables the model to capture both inter-node spatial dependencies and intra-node temporal dynamics, providing a multi-scale fusion of traffic patterns.

### 4.5 PREDICTION HEAD

After feature extraction and fusion via STKAN blocks, the prediction head aggregates temporal information across compressed time steps for each node and applies a linear projection to generate multi-step forecasts. The resulting tensor is then reshaped to match task-specific output dimensions, formally expressed as:

$$\hat{Y}_{t:t+L} = FC_{\text{regression}}(Z_t^i), \tag{16}$$

where $Z_t^i$ denotes the spatio-temporal feature vector of node $i$ at time $t$, and $FC_{\text{regression}}$ is a linear regression layer that maps the encoded features to the predicted values over the future horizon $L$.

## 5 EXPERIMENT

### 5.1 EXPERIMENTAL SETUP

**Datasets.** We evaluate our model on five traffic forecasting datasets, including PEMS04, PEMS07, PEMS08, PEMS-BAY and METR-LA. Following previous work, we divide the PEMS04, PEMS07 and PEMS08 dataset into training, validation, and testsets in a ratio of 6:2:2. For the remaining datasets, we adopt a split ratio of 7:1:2. Detailed statistics of these datasets are shown in A.1.1.

**Baseline.** We compare 11 representative baselines with our proposed STKAN. (i) **Non-spatial modeling-based:** STID Shao et al. (2022), which adopts identity spatio-temporal embeddings and avoids explicit spatial dependency modeling. (ii) **Static spatial-based methods:** STGCN Han et al. (2020), GWNet Wu et al. (2019), AGCRN Bai et al. (2020), GMAN Zheng et al. (2020), MT-GNN Wu et al. (2020) and STDN Cao et al. (2025) combine pre-defined or learned static graph structures with temporal modeling modules. (iii) **Dynamic spatial-based methods:** STAEformer Liu et al. (2023) and STWave Fang et al. (2023) capture time-varying spatial dependencies through adaptive or attention-based mechanisms. (iv) **Spatio-temporal decomposition-based:** StemGNN Cao et al. (2020) and STNorm Deng et al. (2021) decompose spatio-temporal series into separate components for modeling, focusing on disentangling spatial and temporal patterns.

**Evaluation Metrics.** To provide a thorough comparison, we evaluate both the predictive accuracy and computational efficiency of all models. For performance evaluation, we adopt three widely used metrics to quantify the accuracy of traffic forecasting results: Mean Absolute Error (MAE), Root Mean Squared Error (RMSE), and Mean Absolute Percentage Error (MAPE).

### 5.2 PERFORMANCE COMPARISONS

The comprehensive forecasting results are reported in Table 1, where the best outcomes are highlighted in bold and the second-best are underlined. Overall, STKAN consistently achieves superior performance across the five benchmark datasets and three evaluation metrics, with particularly strong advantages on long-term prediction horizons. On flow datasets such as PEMS04 and PEMS08, STKAN yields notable reductions in MAE and RMSE compared to state-of-the-art baselines, demonstrating the effectiveness of its group-wise spatial decomposition and multi-order temporal modeling. On the PEMS-BAY dataset, which is characterized by higher temporal fluctuations, STKAN also shows clear improvements over dynamic attention-based models, confirming its robustness in handling non-stationary traffic dynamics. In contrast, on the more challenging METR-LA dataset, STKAN performs comparably to leading baselines, with narrower margins of improvement due to the highly dynamic and sparse nature of traffic speed signals.

Importantly, beyond accuracy, STKAN provides interpretable spatio-temporal decomposition and kernel-based approximations, offering transparency into spatial group interactions and temporal influence patterns. This interpretability ensures that even when accuracy gains are modest, STKAN delivers unique explanatory power, a quality that is especially valuable in safety-critical applications such as traffic management.

### 5.3 ABLATION STUDY

Table 1: Performance comparisons on normal datasets. We bold the best results and underline the suboptimal results.

| Dataset | | PEMS04 | | | | PEMS07 | | | | PEMS08 | | | | PEMS-BAY | | | | METR-LA | | | |
|---|---|---|---|---|---|---|---|---|---|---|---|---|---|---|---|---|---|---|---|---|---|
| Method | Metric | @3 | @6 | @12 | Avg. | @3 | @6 | @12 | Avg. | @3 | @6 | @12 | Avg. | @3 | @6 | @12 | Avg. | @3 | @6 | @12 | Avg. |
| GWNet (2019) | MAE | 17.89 | 18.80 | 20.35 | 18.81 | 18.71 | 20.14 | 22.35 | 20.10 | 13.67 | 14.59 | 15.99 | 14.58 | 1.31 | 1.65 | 1.99 | 1.59 | 2.69 | 3.08 | 3.52 | 3.04 |
| | RMSE | 28.81 | 30.40 | 32.66 | 30.38 | 30.70 | 33.20 | 36.58 | 33.11 | 21.65 | 23.54 | 25.82 | 23.46 | 2.76 | 3.74 | 4.54 | 3.66 | 5.17 | 6.20 | 7.28 | 6.15 |
| | MAPE | 12.23% | 12.99% | 14.24% | 12.97% | 8.04% | 8.50% | 9.73% | 8.59% | 9.20% | 9.69% | 10.41% | 9.69% | 2.77% | 3.80% | 4.84% | 3.66% | 6.93% | 8.33% | 9.84% | 8.15% |
| STGCN (2020) | MAE | 19.09 | 19.98 | 21.74 | 20.03 | 20.67 | 22.23 | 25.04 | 22.28 | 15.97 | 16.86 | 18.64 | 16.96 | 1.41 | 1.75 | 2.08 | 1.70 | 2.76 | 3.15 | 3.63 | 3.12 |
| | RMSE | 30.1 | 31.57 | 34.07 | 31.63 | 32.76 | 35.71 | 40.4 | 35.83 | 24.56 | 26.29 | 29.00 | 26.38 | 2.93 | 3.89 | 4.69 | 3.81 | 5.30 | 6.32 | 7.47 | 6.29 |
| | MAPE | 12.95% | 13.44% | 14.73% | 13.73% | 8.97% | 9.53% | 10.71% | 9.59% | 10.91% | 11.56% | 12.58% | 11.50% | 3.06% | 3.98% | 4.85% | 3.84% | 7.11% | 8.61% | 10.40% | 8.49% |
| AGCRN (2020) | MAE | 18.55 | 19.50 | 20.77 | 19.45 | 19.29 | 20.82 | 22.81 | 20.74 | 14.78 | 15.96 | 17.63 | 15.91 | 1.35 | 1.69 | 1.96 | 1.61 | 2.87 | 3.24 | 3.63 | 3.19 |
| | RMSE | 29.86 | 31.60 | 33.51 | 31.46 | 31.64 | 34.64 | 38.05 | 34.50 | 22.98 | 25.01 | 27.79 | 25.03 | 2.85 | 3.80 | 4.54 | 3.69 | 5.61 | 6.66 | 7.58 | 6.52 |
| | MAPE | 12.88% | 13.44% | 14.18% | 13.40% | 8.15% | 8.70% | 9.70% | 8.83% | 9.53% | 11.72% | 12.17% | 10.86% | 2.93% | 3.84% | 4.68% | 3.69% | 7.74% | 9.03% | 10.30% | 8.85% |
| StemGNN (2020) | MAE | 19.14 | 20.82 | 24.05 | 21.00 | 20.78 | 23.25 | 27.91 | 23.41 | 14.63 | 16.05 | 18.76 | 16.20 | 1.39 | 1.78 | 2.20 | 1.73 | 2.97 | 3.50 | 4.24 | 3.49 |
| | RMSE | 30.38 | 32.78 | 37.09 | 33.06 | 32.78 | 36.56 | 43.05 | 36.89 | 22.98 | 25.42 | 29.45 | 25.62 | 2.92 | 3.95 | 4.94 | 3.90 | 5.82 | 7.04 | 8.59 | 7.06 |
| | MAPE | 13.68% | 14.82% | 17.44% | 15.05% | 9.29% | 10.21% | 12.45% | 10.39% | 9.28% | 10.50% | 12.26% | 10.55% | 2.94% | 4.09% | 5.32% | 3.97% | 7.97% | 10.06% | 13.01% | 10.04% |
| GMAN (2020) | MAE | 18.23 | 18.78 | 20.12 | 18.81 | 19.31 | 20.41 | 22.20 | 20.48 | 13.76 | 14.59 | 15.83 | 14.81 | 1.35 | 1.66 | 1.93 | 1.58 | 2.81 | 3.15 | 3.49 | 3.07 |
| | RMSE | 29.38 | 30.91 | 31.25 | 30.99 | 31.25 | 33.32 | 36.51 | 33.40 | 22.78 | 24.15 | 26.49 | 24.23 | 2.92 | 3.84 | 4.51 | 3.69 | 5.56 | 6.50 | 7.36 | 6.43 |
| | MAPE | 12.71% | 13.27% | 13.41% | 13.22% | 8.22% | 8.71% | 9.44% | 8.65% | 9.40% | 9.53% | 10.56% | 9.71% | 2.88% | 3.75% | 4.54% | 3.69% | 7.42% | 8.75% | 10.11% | 8.65% |
| MTGNN (2020) | MAE | 18.29 | 19.12 | 20.57 | 19.12 | 19.52 | 21.11 | 23.87 | 21.16 | 14.23 | 15.30 | 16.97 | 15.31 | 1.32 | 1.65 | 1.95 | 1.59 | 2.70 | 3.07 | 3.53 | 3.04 |
| | RMSE | 29.82 | 31.34 | 33.57 | 31.28 | 31.37 | 34.19 | 38.46 | 34.26 | 22.38 | 24.33 | 26.78 | 24.25 | 2.78 | 3.73 | 4.50 | 3.65 | 5.21 | 6.17 | 7.24 | 6.14 |
| | MAPE | 12.62% | 13.09% | 14.31% | 13.14% | 8.77% | 9.10% | 10.34% | 9.27% | 9.42% | 10.57% | 12.17% | 10.60% | 2.75% | 3.68% | 4.55% | 3.53% | 6.85% | 8.17% | 9.81% | 8.08% |
| STNorm (2021) | MAE | 18.30 | 19.12 | 20.27 | 19.05 | 19.21 | 20.57 | 22.66 | 20.51 | 14.48 | 15.45 | 17.03 | 15.45 | 1.33 | 1.66 | 1.96 | 1.58 | 2.80 | 3.18 | 3.56 | 3.12 |
| | RMSE | 29.82 | 31.52 | 33.22 | 31.28 | 31.65 | 34.66 | 38.30 | 34.48 | 23.05 | 25.38 | 27.93 | 25.22 | 2.85 | 3.81 | 4.56 | 3.67 | 5.49 | 6.52 | 7.47 | 6.41 |
| | MAPE | 12.32% | 12.83% | 13.69% | 12.81% | 8.29% | 8.69% | 9.61% | 8.70% | 9.27% | 9.79% | 10.90% | 9.88% | 2.85% | 3.77% | 4.63% | 3.59% | 7.44% | 8.89% | 10.26% | 8.65% |
| STID (2022) | MAE | 17.62 | 18.40 | 19.72 | 18.41 | 18.40 | 19.66 | 21.54 | 19.62 | 13.29 | 14.22 | 15.55 | 14.20 | 1.31 | 1.64 | 1.91 | 1.56 | 2.79 | 3.17 | 3.54 | 3.11 |
| | RMSE | 28.61 | 29.95 | 31.93 | 29.93 | 30.45 | 32.82 | 36.04 | 32.75 | 21.53 | 23.40 | 25.72 | 23.34 | 2.77 | 3.73 | 4.52 | 3.55 | 5.82 | 6.57 | 7.53 | 6.47 |
| | MAPE | 11.95% | 12.42% | 13.50% | 12.51% | 7.77% | 8.28% | 9.22% | 8.31% | 8.65% | 9.29% | 10.32% | 9.31% | 2.77% | 3.73% | 4.52% | 3.55% | 7.66% | 9.27% | 10.77% | 9.01% |
| STAEformer (2023) | MAE | 17.48 | 18.24 | 19.30 | 18.19 | 18.00 | 19.40 | 21.42 | 19.33 | 12.71 | 13.55 | 14.84 | 13.55 | 1.30 | 1.61 | 1.87 | 1.54 | 2.65 | 2.96 | 3.33 | 2.93 |
| | RMSE | 28.89 | 30.31 | 31.99 | 30.18 | 30.42 | 33.30 | 37.02 | 33.21 | 21.63 | 23.48 | 25.80 | 23.44 | 2.77 | 3.68 | 4.34 | 3.57 | 5.11 | 6.01 | 7.02 | 5.98 |
| | MAPE | 11.78% | 12.21% | 13.00% | 12.25% | 7.61% | 8.19% | 9.03% | 8.14% | 8.33% | 8.92% | 9.85% | 8.90% | 2.74% | 3.63% | 4.41% | 3.46% | 6.90% | 8.20% | 9.77% | 8.10% |
| STWave (2023) | MAE | 17.57 | 18.17 | 19.42 | 18.25 | 18.57 | 19.91 | 21.75 | 19.93 | 12.78 | 13.76 | 14.86 | 13.69 | 1.32 | 1.63 | 1.89 | 1.56 | 2.83 | 3.22 | 3.58 | 3.15 |
| | RMSE | 28.88 | 29.95 | 31.78 | 29.99 | 31.59 | 34.36 | 37.35 | 34.09 | 21.59 | 23.79 | 25.77 | 23.57 | 2.80 | 3.71 | 4.35 | 3.59 | 5.34 | 6.71 | 7.60 | 6.56 |
| | MAPE | 11.65% | 12.02% | 13.13% | 12.16% | 7.63% | 8.17% | 9.07% | 8.20% | 8.63% | 9.16% | 10.03% | 9.10% | 2.76% | 3.66% | 4.44% | 3.50% | 7.72% | 9.49% | 11.03% | 9.20% |
| STDN (2025) | MAE | 18.15 | 18.89 | 20.14 | 18.92 | 19.92 | 21.16 | 23.51 | 21.29 | 13.85 | 14.43 | 15.71 | 14.53 | 1.38 | 1.66 | 1.93 | 1.61 | 2.79 | 3.15 | 3.53 | 3.10 |
| | RMSE | 33.14 | 34.64 | 35.85 | 34.33 | 33.56 | 35.88 | 39.61 | 36.02 | 22.31 | 23.90 | 26.21 | 23.96 | 2.95 | 3.83 | 4.47 | 3.66 | 5.59 | 6.61 | 7.56 | 6.51 |
| | MAPE | 19.34% | 19.24% | 19.80% | 19.22% | 12.73% | 12.12% | 14.78% | 12.85% | 12.45% | 11.64% | 10.92% | 11.43% | 3.03% | 3.81% | 4.47% | 3.66% | 7.61% | 9.08% | 10.73% | 8.93% |
| STKAN(Ours) | MAE | **17.40** | **18.13** | **19.15** | **18.09** | **17.94** | **19.27** | **20.97** | **19.16** | **12.62** | **13.48** | **14.80** | **13.45** | **1.29** | **1.61** | **1.87** | **1.54** | **2.69** | **2.99** | **3.39** | **2.97** |
| | RMSE | **28.63** | **30.00** | **31.59** | **29.87** | **30.13** | **32.96** | **36.12** | **32.74** | **21.25** | **23.24** | **25.56** | **23.17** | **2.73** | **3.68** | **4.33** | **3.56** | **5.13** | **6.08** | **7.14** | **6.06** |
| | MAPE | 11.78% | 12.24% | **12.95%** | 12.23% | **7.60%** | **8.05%** | **8.93%** | **8.07%** | **8.28%** | **8.90%** | **9.84%** | **8.88%** | **2.70%** | **3.62%** | **4.35%** | **3.44%** | 6.98% | **8.25%** | **9.71%** | **8.11%** |

**Effectiveness of KAN Modules.** To systematically evaluate the role of KAN components within STKAN, we design three model variants: a) **MLPs**: replacing all KAN modules with standard MLP layers to examine the benefit of functional approximation; b) **w/o SKAN**: removing the spatial block while retaining TKAN to test the importance of inter-node mixing; c) **w/o TKAN**: removing the temporal block while preserving spatial modeling to assess the role of temporal dynamics. As shown in Fig. 3, the full STKAN consistently achieves the best performance across both PEMS04 and PEMS08. Replacing KAN modules with MLPs causes the most significant degradation, confirming that the functional representation capacity of KANs is

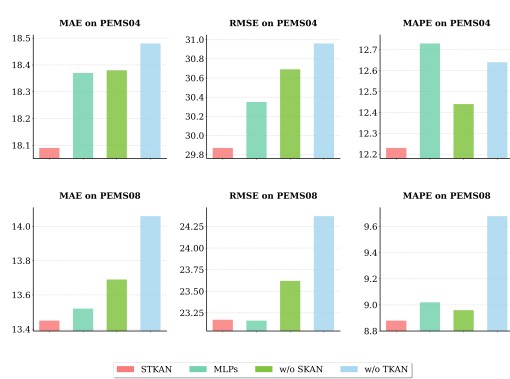

**Figure 3:** KAN modules ablation on PEMS04 and PEMS08.

critical for capturing complex spatio-temporal patterns. Removing either SKAN or TKAN also results in noticeable performance drops, illustrating their complementary roles in modeling spatial and temporal dependencies. Interestingly, the degree of degradation differs across datasets: in PEMS04, removing SKAN or TKAN yields comparable losses due to its dense spatial structure, while in PEMS08, the absence of TKAN has a stronger impact, reflecting the dominance of temporal patterns in simpler networks.

**Effectiveness of Attention Mechanisms.** To evaluate the role of attention in STKAN, we conduct ablation studies on PEMS04 and PEMS08 with the following variants: (1) **w/o S-Attention**: disabling spatial attention while preserving temporal modeling; (2) **w/o T-Attention**: removing temporal attention while keeping spatial modeling; (3) **w/o ST-Attention**: removing both atten-

Table 2: Ablation study of the attention block.

| Dataset | PEMS04 | | | PEMS08 | | |
|---|---|---|---|---|---|---|
| Metric | MAE | RMSE | MAPE | MAE | RMSE | MAPE |
| w/o S-Attention | 18.24 | 30.13 | 12.43% | 13.70 | 23.68 | 9.06% |
| w/o T-Attention | 18.28 | 30.27 | 12.40% | 13.84 | 23.39 | 9.20% |
| w/o ST-Attention | 18.31 | 30.05 | 12.52% | 13.57 | 23.30 | 8.99% |
| **STKAN** | **18.09** | **29.87** | **12.23%** | **13.45** | **23.17** | **8.88%** |

tion modules, leaving only KAN-based token and channel mixing. As shown in Table 2, removing both modules leads to the largest degradation, underscoring the necessity of jointly modeling spatial and temporal dependencies. Removing only one branch causes moderate drops, reflecting their complementary contributions. Although simple MLP replacements reduce complexity, they fail to capture long-range and dynamic interactions. These findings confirm that attention introduces essential inductive biases for handling heterogeneous spatio-temporal patterns and that both spatial and temporal cues are indispensable for accurate forecasting.

## 5.4 HYPER-PARAMETER STUDY

The effect of varying the group number $G$ on the performance of STKAN is analyzed for both the PEMS04 and PEMS07 datasets in Table 3. We find that the number of groups $G$ plays a crucial role in model performance. Too few groups limit spatial abstraction and fail to capture meaningful patterns, while too many cause oversmoothing and loss of local details. Optimal performance is achieved with a moderate $G$, which balances expressiveness and efficiency. Moreover, the appropriate choice of $G$ generally grows with the dataset size and node density, though not in a strictly linear manner, indicating that group-based modeling should adapt to the underlying spatial scale.

Table 3: Performance under varying $G$ values on PEMS04 and PEMS07 datasets.

| Dataset | PEMS04 | | | | | PEMS07 | | | | |
|---|---|---|---|---|---|---|---|---|---|---|
| $G$ | 8 | 12 | 16 | 20 | 24 | 12 | 16 | 20 | 24 | 28 |
| MAE | 18.28 | 18.20 | **18.09** | 18.20 | 18.25 | 19.41 | 19.18 | **19.16** | 19.26 | 19.24 |
| RMSE | 30.76 | 30.16 | **29.87** | 30.00 | 30.25 | 33.18 | 32.77 | **32.74** | 32.86 | 32.93 |
| MAPE | 12.55% | 12.40% | **12.23%** | 12.44% | 12.61% | 8.19% | 8.10% | **8.08%** | 8.17% | 8.13% |

## 5.5 CASE STUDY

We examine the interpretability of the learned spatial structure on the PEMS-BAY network with 325 monitoring nodes. Figure 4 juxtaposes two views: the left panel shows the soft node-to-group assignment matrix, and the right panel overlays the corresponding hard assignments on a light road basemap. Clear banded patterns emerge in the matrix, indicating confident memberships for a subset of groups. In particular, groups such as G8 and G11 display concentrated activations over contiguous columns, while several smaller groups maintain near-uniform probabilities. The map reveals that these dominant groups align with corridor-shaped regions of the freeway network: sensors positioned along the same arterial tend to share the same label (circled areas), whereas junctions and boundary segments exhibit mixed or lower-confidence assignments. Taken together, the two views show that the model organizes nodes into semantically coherent subregions—large groups capture major traffic corridors, smaller groups specialize in localized areas—and provides a transparent account of how spatial information is shared across nodes during forecasting.

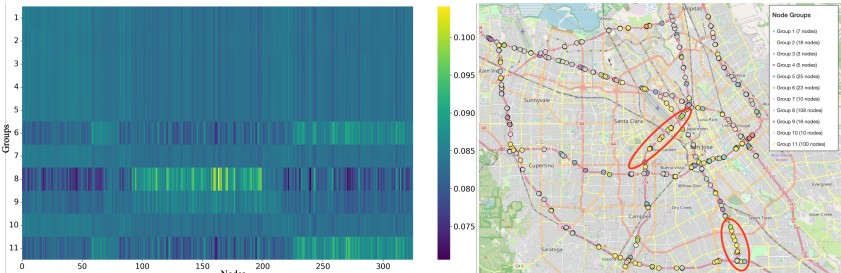

**Figure 4:** Adaptive grouping matrices visualization on PEMS-BAY.

## 6 CONCLUSION

In this paper, we introduced STKAN, a novel interpretable decomposition learning framework for spatio-temporal forecasting. Motivated by the complex nature of real-world traffic dynamics, STKAN employs KANs to disentangle and model spatial and temporal dependencies. Through specialized multi-order modules and an adaptive node-group assignment mechanism, STKAN effectively balances information densities between spatial structures and temporal patterns, reducing inaccuracies in feature interactions. Extensive experiments on benchmark datasets demonstrate that STKAN surpasses state-of-the-art methods in accuracy, interpretability, and scalability. By explicitly modeling critical spatial groupings and temporal patterns, STKAN provides clearer insights into spatio-temporal dynamics, supporting intelligent transportation applications. Future work will focus on refining modeling techniques and enhancing adaptive mechanisms for optimal feature selection, further advancing forecasting in complex spatio-temporal environments.

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

## A APPENDIX

### A.1 IMPLEMENTATION DETAILS

This section provides a comprehensive overview of the implementation setup, including datasets, evaluation metrics, hyperparameters and implementation details.

#### A.1.1 DATASETS

As shown in Table 4, PEMS-BAY is collected by Caltrans' Performance Measurement System (PeMS), whereas METR-LA contains speed readings from loop detectors on Los Angeles County freeways and was curated with the DCRNN release Li et al. (2017). The PEMS04/07/08 datasets are PeMS-based traffic flow benchmarks with 5-minute aggregation.

Table 4: Summary of Five Spatio-temporal Benchmarks

| Dataset | Category | Sensors | Time Steps | Time Interval | Time Span (Y/M/D) |
|---------|----------|---------|------------|---------------|-------------------|
| PEMS04 | Traffic flow | 307 | 16992 | 5 min | 2018/01/01 – 2018/02/28 |
| PEMS07 | Traffic flow | 883 | 28224 | 5 min | 2017/05/01 – 2017/08/31 |
| PEMS08 | Traffic flow | 170 | 17856 | 5 min | 2016/07/01 – 2016/08/31 |
| PEMS-BAY | Traffic speed | 325 | 52116 | 5 min | 2017/01/01 – 2017/06/30 |
| METR-LA | Traffic speed | 207 | 34272 | 5 min | 2012/03/01 – 2012/06/27 |

#### A.1.2 MODEL ARCHITECTURE

In the experiments, we use the traffic flow of the last 12 time steps to predict the traffic flow of the next 12 time steps, and record the prediction performance of the 3rd, 6th, 12th steps and the average. The dimension of hidden representations in our model is set as 128 and the dimension of embedding layer is set as 32. We set the Adam optimizer with an initial learning rate of 0.002, where the learning rate follows a step-wise decay strategy, and the batch size is set as 64. During the training phase, we employ the early stopping strategy with tolerance 30 for 200 epochs. All experiments are conducted using PyTorch on NVIDIA RTX H100 GPU with 60GB of memory.

### A.2 EVALUATION METRICS

We adopt three commonly used regression metrics, namely Mean Absolute Error (MAE), Root Mean Squared Error (RMSE), and Mean Absolute Percentage Error (MAPE), to evaluate the prediction performance. Suppose the ground truth spatio-temporal data is denoted as $Y = \{y_1, y_2, \ldots, y_N\}$, and the corresponding predicted values are $\hat{Y} = \{\hat{y}_1, \hat{y}_2, \ldots, \hat{y}_N\}$, where $N$ is the number of total testing samples. The three metrics can be formulated as follows:

$$\text{MAE} = \frac{1}{N} \sum_{i=1}^{N} |y_i - \hat{y}_i| \tag{17}$$

MAE measures the average absolute difference between the predicted values and the actual values.

$$\text{RMSE} = \sqrt{\frac{1}{N} \sum_{i=1}^{N} (y_i - \hat{y}_i)^2} \tag{18}$$

RMSE penalizes larger errors more heavily, making it more sensitive to outliers.

$$\text{MAPE} = \frac{100\%}{N} \sum_{i=1}^{N} \left| \frac{y_i - \hat{y}_i}{y_i} \right| \tag{19}$$

MAPE expresses the prediction error as a percentage, which provides an intuitive measure of prediction accuracy.

### A.3    BASELINE

The following spatio-temporal models are implemented using the BasicTS framework (`https://github.com/GestaltCogTeam/BasicTS`) to ensure consistency in preprocessing, training, and evaluation:

**Graph WaveNet** Wu et al. (2019): A graph neural network that adaptively learns spatial dependencies and captures long-range temporal patterns with dilated convolutions.

**STGCN** Han et al. (2020): A graph convolutional network that efficiently captures spatio-temporal correlations in traffic data.

**AGCRN** Bai et al. (2020): A recurrent graph model with adaptive modules that learns node-specific patterns and infers spatial dependencies without pre-defined graphs.

**StemGNN** Cao et al. (2020): A spectral-domain model that jointly captures inter-series and temporal dependencies using Graph and Discrete Fourier Transforms.

**GMAN** Zheng et al. (2020): An encoder–decoder model with spatio-temporal and transform attention to capture relations between historical and future traffic states.

**MTGNN** Wu et al. (2020): A graph neural network with adaptive graph learning and mix-hop propagation to capture latent spatial and temporal dependencies in multivariate time series.

**STNorm** Deng et al. (2021): A normalization-based approach that refines temporal and spatial components to improve multivariate time series forecasting.

**STID** Shao et al. (2022): A simple MLP-based model that incorporates spatial and temporal identity information to improve efficiency and accuracy in multivariate time series forecasting.

**STAEformer** Liu et al. (2023): A transformer-based model with spatio-temporal adaptive embedding to effectively capture intrinsic traffic patterns.

**STWave** Fang et al. (2023): A disentangle-fusion framework that decouples traffic data into trends and events, modeling them with dual spatio-temporal networks to handle distribution shifts.

**STDN** Cao et al. (2025): A dynamic graph model with spatio-temporal embeddings and trend–seasonality decomposition to capture complex traffic dynamics.

### A.4    VISUALIZATION

To further examine the interpretability of KAN in the group space, we analyze several representative output groups. Figure 5 plots, over the standardized input domain, each group's final response (black solid curve) together with a linear reconstruction using a small number of KAN basis functions weighted by their learned coefficients (colored dashed curves). The visualization reveals clear

functional specialization across groups. Low-variation groups (e.g., Group 15, Group 4, Group 2) exhibit near-linear or weak-curvature monotonic mappings whose final curves are well explained by only two to three basis functions, indicating that these groups primarily implement smoothing/low-pass behavior suitable for corridors or peripheral subnetworks with modest fluctuations. In contrast, complex nonlinear groups (e.g., Group 14) display pronounced asymmetric curvature and strong nonlinearity, requiring multiple basis functions to capture their shape, suggesting that the model concentrates nonlinear capacity on traffic regions with complex dynamics such as junctions or bottlenecks. This "few atoms explain the final function" view both exposes a functional prototype for each output group (low-pass, monotone, or strongly nonlinear) and demonstrates that KAN achieves task-aligned, interpretable capacity allocation in group space: simple regions are handled by simple functions, whereas complex regions are assigned richer nonlinear function families.

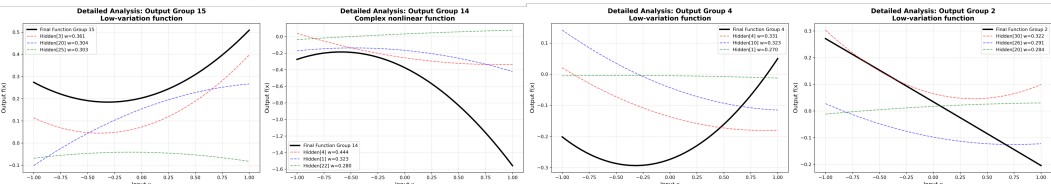

**Figure 5:** Symbolization of the group function.

## B USE OF LARGE LANGUAGE MODELS (LLMs)

During the preparation of this manuscript, we used a Large Language Model (LLM) solely for language polishing purposes. Specifically, the LLM was applied to improve grammar, clarity, and readability of sentences drafted by the authors. No part of the research ideation, experiment design, data analysis, or substantive content generation was conducted by the LLM. The scientific contributions, arguments, and conclusions presented in this paper are entirely the work and responsibility of the authors. The authors take full responsibility for all contents of the paper, including sections where the LLM-assisted refinements were applied.

