# OpenReview forum: "STKAN: Kolmogorov-Arnold Networks for Spatio-Temporal Forecasting"
_ICLR.cc/2026/Conference — ICLR 2026 Conference Withdrawn Submission_

### Official Review · Reviewer_rD34 · 2025-10-16

**Soundness:** 3
**Presentation:** 2
**Contribution:** 2
**Rating:** 2
**Confidence:** 5

**Summary:**

The paper proposes STKAN, a Kolmogorov–Arnold Network-based framework for spatio-temporal forecasting. Motivated by the heterogeneity between spatial and temporal information densities in traffic data, the authors design separate spatial and temporal KAN modules that model these dependencies independently before fusing them for prediction. The architecture integrates learnable spatial grouping, multi-order Taylor-based KAN layers, and Transformer-style attention for global fusion. Experiments on standard traffic datasets  show modest yet consistent performance gains over recent baselines. The model also claims interpretability through spatial grouping visualization and basis-function analysis.

**Strengths:**

1. The paper is technically sound overall, with a detailed methodological section and ablation studies examining the contributions of spatial/temporal KAN blocks and attention modules.

2. The presentation is mostly clear and structured, though occasionally verbose. Figures and equations are appropriately referenced. The interpretability section with group visualizations is a nice touch.

3. While improvements are moderate, the decomposition view and KAN formulation could encourage future exploration of nonstandard activation structures in spatio-temporal learning.

**Weaknesses:**

1. The authors introduce the concept of varying information density between spatial and temporal patterns but provide neither a clear formal definition nor an intuitive explanation within their modeling context. It remains unclear how this concept concretely motivates the design of STKAN.

2. The claim that MLP-based models are limited due to fixed activation functions is questionable. The universal approximation theorem already ensures the expressiveness of MLPs; thus, the motivation for replacing them with KANs needs stronger theoretical or empirical justification.

3. The role of decomposition via KAN in understanding spatial and temporal correlations is not clearly articulated. The paper lacks a discussion of why KANs, compared with other decomposition approaches , are particularly advantageous.
[1] Wang, Shiyu, et al. "Timemixer: Decomposable multiscale mixing for time series forecasting." arXiv preprint arXiv:2405.14616 (2024).
[2] Deng, Jinliang, et al. "Disentangling structured components: Towards adaptive, interpretable and scalable time series forecasting." IEEE Transactions on Knowledge and Data Engineering 36.8 (2024): 3783-3800.
[3] Hu, Yifan, et al. "Adaptive multi-scale decomposition framework for time series forecasting." Proceedings of the AAAI Conference on Artificial Intelligence. Vol. 39. No. 16. 2025.


4. Several relevant prior studies applying KAN to time-series forecasting are not properly cited or discussed, reducing the completeness of the related work.
[1] Vaca-Rubio, Cristian J., et al. "Kolmogorov-arnold networks (kans) for time series analysis." arXiv preprint arXiv:2405.08790 (2024).
[2] Huang, Songtao, et al. "Timekan: Kan-based frequency decomposition learning architecture for long-term time series forecasting." arXiv preprint arXiv:2502.06910 (2025).
[3] Han, Xiao, et al. "Are KANs Effective for Multivariate Time Series Forecasting?." arXiv preprint arXiv:2408.11306 (2024).

5. The reported performance improvements over state-of-the-art baselines (e.g., STAEformer) are marginal and not always consistent with the original results reported in those papers. The superiority of STKAN appears incremental rather than substantial.

6. The code is not released, which limits reproducibility and transparency of the experimental claims.

**Questions:**

Please refer to the weaknesses.

---

### Official Review · Reviewer_9TyE · 2025-10-31

**Soundness:** 2
**Presentation:** 3
**Contribution:** 2
**Rating:** 4
**Confidence:** 5

**Summary:**

The paper introduces a novel spatio-temporal forecasting framework STKAN that leverages the expressive power of Kolmogorov–Arnold Networks to disentangle spatial and temporal dependencies in complex traffic data. Unlike prior spatio-temporal models that jointly encode both dimensions, STKAN explicitly separates and models them through specialized spatial (SKAN) and temporal (TKAN) modules. These modules employ multi-order KAN layers with learnable nonlinear functions, allowing the network to flexibly adapt to varying spatial structures and temporal dynamics. Experiments on five widely used benchmark datasets show that STKAN consistently achieves state-of-the-art performance.

**Strengths:**

S1. The paper makes a conceptually novel contribution by being among the first to apply Kolmogorov–Arnold Networks (KANs) to the spatio-temporal forecasting domain. While KANs have been explored in time-series or symbolic regression contexts, their integration into spatio-temporal modeling—particularly for traffic forecasting—is new and meaningful.

S2. The use of multi-order Taylor-based KAN layers allows the model to flexibly approximate complex nonlinear relationships across spatial and temporal dimensions.

S3. STKAN achieves state-of-the-art results across five widely used benchmark datasets, outperforming both traditional graph-based and modern transformer-based baselines in most settings.

**Weaknesses:**

W1. While the paper claims that KANs enable disentangled spatial–temporal modeling, the motivation behind this choice is not clearly justified or theoretically supported. The Kolmogorov–Arnold theorem describes the universal decomposition of multivariate functions into sums of univariate ones, but the paper does not convincingly explain why or how KAN’s learnable functional mappings naturally facilitate spatial–temporal separation in traffic data. Moreover, many prior architectures (e.g., STAEFormer, GMAN) already use isolated spatial and temporal attention blocks to decouple spatial and temporal dependencies without invoking KANs. The authors do not provide sufficient argumentation or empirical evidence to show that KAN-based decomposition offers a fundamentally superior mechanism compared to attention-based or graph convolution-based disentanglement.

W2. Although the paper includes several ablation experiments, it omits critical configurations that are essential to understanding the contribution of each KAN component. Specifically, the paper does not include w/o S-KAN and w/o T-KAN. Without these key ablations, it remains unclear whether the performance improvement comes from the KAN architecture itself or from other design choices (e.g., the attention-based fusion, adaptive grouping).

W3. The paper does not include any efficiency or computational complexity analysis, which is crucial for assessing practical applicability.

W4. Although the paper claims that STKAN “explicitly decomposes” spatial and temporal dependencies, it does not include qualitative case studies or visual evidence showing how this decomposition manifests in practice.

**Questions:**

See weaknesses.

---

### Official Review · Reviewer_Z5b3 · 2025-11-01

**Soundness:** 4
**Presentation:** 4
**Contribution:** 1
**Rating:** 4
**Confidence:** 2

**Summary:**

The paper proposes temporal and spatial KANs for spatiotemporal forecasting. The model is validated on several traffic datasets.

**Strengths:**

- Clarity: The paper is easy to read, and the reader's understanding is supported with intuitive figures (e.g. Figure 2) that show information flow. The authors also provide appropriate background and preliminaries to help the reader understand KANs.
- Quality: The authors ran extensive experiments, including benchmarking on 5 datasets, ablation studies, and hyperparameter studies.

**Weaknesses:**

- Originality: There have been other works that apply KANs to traffic prediction. The author should include these in the related works to differentiate this work from others. E.g.
  - Zhao, W., Yuan, G., Bing, R., Liu, X., & Zhang, G. (2024, December). GraphKAN: An Efficient Graph Kolmogorov Arnold Networks for Traffic Forecasting. In 2024 IEEE International Conference on Smart City (SmartCity) (pp. 23-30). IEEE.
  - Chen, Y., Li, S., Zhao, N., Zheng, R., & Li, Y. (2024, October). BiLSTM-KAN: A Time Series-based Traffic Flow Forecasting Model. In Proceedings of the 2024 13th International Conference on Computing and Pattern Recognition (pp. 314-319).

**Questions:**

- Are the baseline results in the paper referenced from the original work? If so, is the data preprocessing/loading pipeline the same?
- What is the training time for the model? What about memory size?
- How is STKAN different from other KAN based spatiotemporal forecasting methods?

---

### Official Review · Reviewer_44fP · 2025-11-02

**Soundness:** 3
**Presentation:** 3
**Contribution:** 2
**Rating:** 4
**Confidence:** 4

**Summary:**

This article argues that previous studies, which simply connect spatial and temporal features or fuse them with fixed nonlinear methods, usually fail to reconcile the differing information densities and statistical properties of the two dimensions. Based on this issue, it proposes designing a framework that decomposes and models spatio-temporal patterns separately to improve representation quality and predictive performance. The specific approach is to use KAN to unravel and model spatio-temporal dependencies. Through specialized multi-order modules and an adaptive node group assignment mechanism, STKAN effectively balances the information density between spatial structures and temporal patterns, reducing inaccuracies in feature interactions.

**Strengths:**

S1: This paper is well-written and easy to follow.
S2: The experimental results seem to be good compared with baseline methods, the baseline methods are comprehensive and most of these are published in recent years.

**Weaknesses:**

W1: The article points out that previous studies, which simply connect spatial and temporal features or fuse them with fixed nonlinear methods, typically fail to reconcile the differing information densities and statistical properties of the two dimensions. The term 'information density' is mentioned multiple times in the article, but no clear definition is provided. Therefore, the statement regarding the inability to reconcile differing information densities in two dimensions is rather vague, making it unclear what the consequences of previous methods were and what specific problem the article aims to solve.

W2: Spatiotemporal KAN for traffic or time-series prediction is not so novel idea in the spatiotemporal data mining community, considering the following studies, the technical contribution of this paper seems to be incremental.
[1] Zhao, W., Yuan, G., Zhang, Y., Liu, X., Liu, S., & Zhang, L. (2026). An interpretable and efficient multi-scale spatio-temporal neural network for traffic flow forecasting. Expert Systems with Applications, 296, 128961.
[2] Zhang, W., Xia, D., Chang, G., Zheng, Z., Hu, Y., Huo, Y., ... & Li, H. St-Kan-Former: A Novel Spatiotemporal Transformer Neural Network for Air Quality Prediction.
[3] Chen, Y., Li, S., Zhao, N., Zheng, R., & Li, Y. (2024, October). BiLSTM-KAN: A Time Series-based Traffic Flow Forecasting Model. In Proceedings of the 2024 13th International Conference on Computing and Pattern Recognition (pp. 314-319).
[4] Genet, R., & Inzirillo, H. (2024). A temporal kolmogorov-arnold transformer for time series forecasting. arXiv preprint arXiv:2406.02486.

W3: In Appendix Figure 5, the second figure does not appear significantly different from the other three. The article explains that it corresponds to different groups, but further supporting evidence is needed.

**Questions:**

Q1: Technical contribution and novelties considering W2.
Q2: The article visualizes the adaptive node group assignment matrix and the corresponding sensor map in Figure 4, interpreting it as corridors and intersections belonging to different groups. The method described in the paper involves linearly transforming the adaptive node group assignment matrix along the channel dimension to obtain embedded vectors. In my view, the banding in the figure is due to sensors that are geographically close having similar embeddings, for example, sensors numbered 250-300 exhibit similar embeddings. Does the article conclude that sensors numbered 250-300 belong to both groups 6 and 11? Is the explanation for concluding that these sensors belong to different groups reasonable based on the above operation?
Q3: A more clearer motivation of the proposed method considering W1.

---

### Note · Authors · 2026-01-12

I have read and agree with the venue's withdrawal policy on behalf of myself and my co-authors.